# On the Gaussian distribution of the Mann-Kendall tau in the case of autocorrelated data

Tristan Gamot[1,2☯*], Nils Thibeau--Sutre[1,3☯*], Tom J. M. Van Dooren[1]

**1** Sorbonne Université (SU), Université Paris Cité (UPC), Université Paris Est Créteil (UPEC), CNRS, IRD, INRAE, Institut d'écologie et des sciences de l'environnement de Paris, IEES Paris, Paris, France, **2** Centre de Physique Théorique (CPHT), CNRS, École polytechnique, Institut Polytechnique de Paris, Palaiseau, France, **3** Aix Marseille Univ, CNRS, I2M, Marseille, France

☯ These authors contributed equally to this work.
* tristan.gamot@ens-paris-saclay.fr (TG); nils.thibeau-sutre@univ-amu.fr (NT–S)

## Abstract

Non-parametric Mann-Kendall tests for autocorrelated data rely on the assumption that the distribution of the normalized Mann-Kendall tau is Gaussian. While this assumption holds asymptotically for stationary autoregressive processes of order 1 (AR(1)) and simple moving average (SMA) processes when sampling over an increasingly long period, it often fails for finite-length time series. In such cases, the empirical distribution of the Mann-Kendall tau deviates significantly from the Gaussian distribution. To assess the validity of this assumption, we explore an alternative asymptotic framework for AR(1) and SMA processes. We prove that, along upsampling sequences, the distribution of the normalized Mann-Kendall tau does not converge to a Gaussian but instead to a bounded distribution with strictly positive variance. This asymptotic behavior suggests scaling laws which determine the conditions under which the Gaussian approximation remains valid for finite-length time series generated by stationary AR(1) and SMA processes. Using Shapiro-Wilk tests, we numerically confirm the departure from normality and establish simple, practical criteria for assessing the validity of the Gaussian assumption, which depend on both the autocorrelation structure and the series length. Finally, we illustrate these findings with examples from existing studies.

## 1 Introduction

The Mann-Kendall test is a non-parametric statistical method designed to assess whether a time series exhibits a monotonic trend, based on the Mann-Kendall tau statistic. Initially introduced by Henry B. Mann [1] and later refined by Maurice G. Kendall [2], the test leverages the fact that for independent and identically distributed (i.i.d.) data, the distribution of the Mann-Kendall tau normalized by its variance converges asymptotically to a Gaussian distribution.

To address scenarios where the data are not i.i.d., extensions have been developed for autocorrelated datasets [3,4]. These modifications incorporate adjustments

**Data availability statement:** The code for reproducing figures is accessible at https://doi.org/10.5281/zenodo.18414688.

**Funding:** AAP2023 FIRE mini-grant.

**Competing interests:** The authors have declared that no competing interests exist.

to account for serial correlation; however, they do not establish asymptotic normality and instead proceed under the assumption that this property remains valid.

When considering autoregressive processes of order 1 (AR(1)) and simple moving average processes of order q (SMA(q)) with fixed parameters, classical results on the Central Limit Theorem for *U-statistics* applied to $\alpha$-mixing processes [5] and *m*-dependent processes [6] establish that the normalize's asymptotic distribution is Gaussian. Yet, when the lag-1 autocorrelation parameter or the order of the moving average is high relative to the finite length of a time series, the empirical distribution of the normalized Mann-Kendall statistic is far from Gaussian (see for example [7]). This is a key observation as, in practice, the family of Mann-Kendall tests are applied to autocorrelated time series of finite length and Gaussian distributions are used as an approximation. For instance, modified versions of the original test for autocorrelated data are widely used in hydrological studies that typically involve time series with several dozen to hundreds of data points [3,4,7]. So how can one determine whether the Gaussian approximation is justified? Since the Mann-Kendall tau is based on pairwise comparisons of all random variables in a sequence, this question requires analyzing asymptotic regimes where the density of pairs with non-negligible dependence is non-zero. To explore this, we examine specific sequences of time series of increasing length generated by AR(1) or SMA(q) processes, as defined in Sect 2. The relevance of asymptotic results remains a point of discussion in statistics (e.g. [8]) and exact and Monte Carlo methods have been proposed as general remedies for hypothesis testing on finite samples (e.g. [9,10]). However, in the context of time series, there is scope for different asymptotic results than the ones usually envisaged. Here we will investigate sequences differing from the ones investigated previously where the limits converged to Gaussian distributions. Each time series in the investigated sequences is generated by a stationary process whose parameters depend on the series length and, therefore, on its position within the sequence. For the AR(1) case, this sequence of time series corresponds to refining the sampling of a continuous Ornstein-Uhlenbeck process within a fixed time window - i.e. upsampling - which increases autocorrelation. Regarding the SMA(q) process, these sequences amount to averaging increasingly larger samples of a white noise process while maintaining constant relative window size. Such averages are used to construct statistics given the null model when testing for the presence of critical transitions in time series - also known as early warning signals [11]. In Sect 3, we prove that the asymptotic distribution of the normalized Mann-Kendall tau of these two types of times series cannot be Gaussian. These proofs suggest natural scalings for deciding whether the Gaussian approximation is suitable for time series of finite length generated by AR(1) or SMA(q) processes. Finally, Sect 4 numerically illustrates the departure from Gaussian behavior using the Shapiro-Wilk test for normality and confirms that these scalings are appropriate. We can therefore provide easy-to-implement criteria, for given values of parameters and time series length, to decide whether the Gaussian approximation - and hence the Mann-Kendall tests - may be appropriate or not for a time series.



## 2 Assumptions and examples

Let $X$ be a random variable, and $X_i$, $1 \leq i \leq n$, be $n$ random variables having the same distribution as $X$. The Mann-Kendall tau for the time series $(X_i)_{1 \leq i \leq n}$ is defined as [1,12]:

$$\tau\left((X_i)_{1 \leq i \leq n}\right) := \frac{1}{\binom{n}{2}} \sum_{1 \leq i < j \leq n} A_{ij}, \tag{1}$$

where $A_{ij} := \text{sign}(X_j - X_i) = \pm 1$ and $\binom{n}{2} := \frac{n(n-1)}{2}$.

We assume that the distribution of $X$ is such that there are no ties. For simplicity, we denote the Mann-Kendall tau for the sequence $(X_i)_{1 \leq i \leq n}$ by $\tau_n$. This non-parametric statistic is a special case of Kendall's rank correlation coefficient and is used for detecting monotonic trends. It ranges from -1 (strictly decreasing trend) to +1 (strictly increasing trend).

If the $X_i$ are independent, Kendall [2] proved that:

$$\frac{\tau_n}{\sqrt{\mathbb{V}(\tau_n)}} \underset{n \to \infty}{\sim} \sqrt{\frac{9n}{4}} \tau_n \xrightarrow[n \to \infty]{d} \mathcal{N}(0,1), \tag{2}$$

where $\mathbb{V}(\tau_n)$ is the variance of the random variable $\tau_n$, $\mathcal{N}(0,1)$ is the standard Gaussian random distribution and $\xrightarrow[n \to \infty]{d}$ stands for convergence in distribution. Let us stress that it is the normalized random variable $\frac{\tau_n}{\sqrt{\mathbb{V}(\tau_n)}}$ which converges in distribution towards a Gaussian and not $\tau_n$ (which is bounded between $-1$ and $+1$).

Let us now consider the case where the $X_i$ are not independent and are Gaussian random variables. The variance of the Mann-Kendall tau simplifies as follows using Greiner's equality [13]:

$$\mathbb{V}(\tau_n) = \frac{1}{\binom{n}{2}^2} \sum_{1 \leq i < j \leq n} \sum_{1 \leq k < l \leq n} \mathbb{E}(A_{ij} A_{kl}) \tag{3}$$

$$= \frac{1}{\binom{n}{2}^2} \sum_{1 \leq i < j \leq n} \sum_{1 \leq k < l \leq n} \frac{2}{\pi} \arcsin\left(\text{corr}\left(X_j - X_i, X_l - X_k\right)\right), \tag{4}$$

where $\mathbb{E}(A_{ij} A_{kl})$ is the expectation of the random variable $A_{ij} A_{kl}$ and corr denotes the Pearson correlation coefficient.

In this paper, we only consider sequences $(X_i)_{1 \leq i \leq n}$ of identically distributed Gaussian random variables (for a fixed $n$) that verify the following property:

**Assumption 1** (Correlation function). $\exists \tilde{\rho} : \mathbb{N} \to [-1, 1]$ *such that*, $\forall 1 \leq i, j \leq n$, *the sequence* $(X_i)_{1 \leq i \leq n}$ *satisfies*

$$\text{corr}(X_i, X_j) = \tilde{\rho}(|j - i|). \tag{5}$$

We then call $\tilde{\rho}$ the autocorrelation function of the sequence $(X_i)_{1 \leq i \leq n}$. Note that, if $\forall 1 \leq i, j \leq n$, $\mathbb{E}(X_i) = \mathbb{E}(X_j)$ and $\mathbb{E}(X_i^2) < \infty$, then Assumption 1 means that the sequence is weak-sense (or wide-sense) stationary. In the case of Gaussian random variables, weak stationarity is equivalent to strict stationarity. This condition is satisfied by all examples considered in this article.

In the remainder of this section, we introduce an assumption on the existence of a renormalized asymptotic autocorrelation in a sequence of time series. This enables us to derive an easy-to-handle expression for the asymptotic variance of the Mann-Kendall tau of these time series.

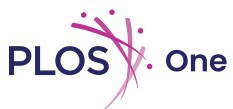

**Assumption 2** (Correlation function renormalization). $\exists \rho : [0, 1] \to [-1, 1]$ *such that,* $\forall 1 \leq i_n, j_n \leq n, \frac{i_n}{n} \to x$ *and* $\frac{j_n}{n} \to y$, *the sequence of sequences* $\left( (X_i^{(n)})_{1 \leq i \leq n} \right)_{n \in \mathbb{N}}$ *satisfies*

$$\lim_{n \to \infty} \mathrm{corr}(X_{i_n}^{(n)}, X_{j_n}^{(n)}) = \rho\left( |x - y| \right), \tag{6}$$

*where, for all positive integers* $n, (X_i^{(n)})_{1 \leq i \leq n}$ *is a sequence of n random variables.*

In this article, we choose sequences of identically distributed Gaussian random variables whose correlation depends on the length $n$ of the sequence, hence the superscript $^{(n)}$. Then, random variables from two sequences $(X_i^{(n)})_{1 \leq i \leq n}$ and $(X_i^{(m)})_{1 \leq i \leq m}$ of length $n$ and $m$ (with $n \neq m$) have different correlation functions. Hereafter we introduce the two classes of sequences that we have in mind for considering Assumptions 1 and 2:

**Example 1** (Autoregressive process of order 1 AR(1)). *Let* $0 < k_{\mathrm{tot}} < 1, n \geq 1, k_n = k_{\mathrm{tot}}^{1/(n-1)}, (X_i^{(n)})_{1 \leq i \leq n}$ *such that:*

$$X_1^{(n)} \sim \mathcal{N}(0, 1), \qquad X_i^{(n)} = k_n X_{i-1}^{(n)} + \epsilon_i^{(n)} \quad \text{if } 2 \leq i \leq n,$$

*where, for a given n, the* $\epsilon_i^{(n)}$ *are independent and identically distributed Gaussian random variables:* $\epsilon_i^{(n)} \sim \mathcal{N}(0, 1 - k_n^2)$.

Then for the sequence of sequences $((X_i^{(n)})_{1 \leq i \leq n})_{n \geq 1}$, Assumptions 1 and 2 are true. In particular, $\rho(x) = k_{\mathrm{tot}}^x$. Note that we only consider $k > 0$ with this renormalisation.

For instance, if $k_{tot} = 10^{-8}$, the first elements of the sequence of sequences are given by:

1. $(X_1^{(1)})$ where $X_1^{(1)} \sim \mathcal{N}(0, 1)$,
2. $(X_1^{(2)}, X_2^{(2)})$ where $X_1^{(2)} \sim \mathcal{N}(0, 1)$ and $X_2^{(2)} = k_2 X_1^{(2)} + \epsilon_2^{(2)}$ where $k_2 = k_{\mathrm{tot}}^{1/(2-1)} = 10^{-8}$,
3. $(X_1^{(3)}, X_2^{(3)}, X_3^{(3)})$ where $X_1^{(3)} \sim \mathcal{N}(0, 1)$, $X_2^{(3)} = k_3 X_1^{(3)} + \epsilon_2^{(3)}$, and $X_3^{(3)} = k_3 X_2^{(3)} + \epsilon_3^{(3)}$, where $k_3 = k_{\mathrm{tot}}^{1/(3-1)} = 10^{-4}$,
4. $(X_1^{(4)}, X_2^{(4)}, X_3^{(4)}, X_4^{(4)})$ where $X_1^{(4)} \sim \mathcal{N}(0, 1)$, $X_2^{(4)} = k_4 X_1^{(4)} + \epsilon_2^{(4)}$, $X_3^{(4)} = k_4 X_2^{(4)} + \epsilon_3^{(4)}$, and $X_4^{(4)} = k_4 X_3^{(4)} + \epsilon_4^{(4)}$, where $k_4 = k_{\mathrm{tot}}^{1/(4-1)} = 2.15 \times 10^{-3}$,
5. *etc.*

Example 1 presents sequences of time series of length $n$ generated by AR(1) processes with increasing autocorrelation at lag-1 $k_n = k_{\mathrm{tot}}^{1/(n-1)}$. The time series are comparable to upsampling an Ornstein–Uhlenbeck process over [0,1] as described in [14] (§5) with the sequence $(X_i^{(n)})_{1 \leq i \leq n}$ representing a regular subdivision of this process with steps of $1/(n-1)$. For example, starting with two values sampled at the edges of the interval, that is at time 0 and 1, the autocorrelation between these is equal to $k_{\mathrm{tot}}$. If one subdivides the interval into $n > 2$ uniformly spaced samples, then the autocorrelation between two successive values has to be $k_{\mathrm{tot}}^{1/(n-1)}$ so that the autocorrelation between the first and last values remains $\left( k_{\mathrm{tot}}^{1/(n-1)} \right)^{n-1} = k_{\mathrm{tot}}$. So this case arises naturally when increasing the sampling of the same experiment of finite duration.

**Example 2** (Simple moving average SMA). *Let* $a > 0, n \geq 1, q_n = \lfloor an \rfloor$, *where* $\lfloor \cdot \rfloor$ *is the floor function, and* $(X_i^{(n)})_{1 \leq i \leq n}$ *such that:*

$$X_i^{(n)} = \sum_{j=1}^{q_n} \epsilon_{i-j}^{(n)}, \quad 1 \leq i \leq n,$$

*where the* $\epsilon_j^{(n)}$ *are independent and identically distributed Gaussian random variables:* $\epsilon_j^{(n)} \sim \mathcal{N}(0, 1)$.

Then for the sequence $((X_i^{(n)})_{1 \leq i \leq n})_{n \geq 1}$, Assumptions 1 and 2 are true. In particular, $\rho(x) = \max(1 - \frac{x}{a}, 0)$.

*A standard parameter characterizing moving average processes is the relative window size. As an example, starting with a dataset of N points and averaging by groups of q contiguous points creates a moving average dataset of length $n = N - q + 1$ with a relative window size $\alpha = \frac{q}{N} = \frac{q}{n+q-1}$. Then, if the window size q depends on n and $q_n/n \xrightarrow[n\to\infty]{} a$, the relative window size is asymptotically $\frac{q_n}{q_n+n-1} \xrightarrow[n\to\infty]{} \frac{a}{a+1}$.*

Example 2 presents a sequence of time series generated by SMA processes whose relative window size tends towards a constant when the length of the time series goes to infinity. This type of time series arises naturally when considering averaging over windows of fixed relative size. For example, when testing for the presence of critical transitions in time series (also known as early warning signals), a methodology involves averaging over a constant fraction of the time series [11].

Finally, following the definition by [15], the Mann-Kendall tau can be defined as a *U-statistic* with a non-symmetric kernel of degree 2 [16]. Classical results on the Central Limit Theorem for *U-statistics* applied to $\alpha$-mixing processes [5] and *m*-dependent processes [6] establish that the asymptotic distribution of the Mann-Kendall tau is Gaussian for AR(1) and MA(q) processes with fixed lag-1 autocorrelation parameter *k* for AR(1) processes and order *q* of SMA processes. However, in Examples 1 and 2, we are considering sequences of time series generated by processes whose parameters depends on the length of the time series. For the example cases, we will prove that the distribution of the normalized Mann-Kendall tau cannot be asymptotically Gaussian.

## 3 Asymptotic variance of the Mann-Kendall tau for renormalized ARMA process

In this section, we delve deeper into the two classes of examples introduced, building upon Assumptions 1 and 2 to derive a key lemma for calculating the variance of the Mann-Kendall tau. All proofs are provided in S1 Appendix.

As before, let's consider a sequence $(X_i^{(n)})_{1 \leq i \leq n}$ of identically distributed Gaussian random variables. Then, under Assumption 1, we obtain:

$$\text{corr}(X_j^{(n)} - X_i^{(n)}, X_k^{(n)} - X_l^{(n)}) = \frac{\text{cov}(X_j^{(n)} - X_i^{(n)}, X_k^{(n)} - X_l^{(n)})}{\sqrt{\mathbb{V}(X_j^{(n)} - X_i^{(n)})}\sqrt{\mathbb{V}(X_l^{(n)} - X_k^{(n)})}} \tag{7}$$

$$= \frac{\tilde{\rho}(|l - j|) - \tilde{\rho}(|l - i|) - \tilde{\rho}(|k - j|) + \tilde{\rho}(|k - i|)}{2\sqrt{1 - \tilde{\rho}(|j - i|)}\sqrt{1 - \tilde{\rho}(|l - k|)}} . \tag{8}$$

**Lemma 1** (Renormalization). *Suppose Assumptions 1 and 2 hold for a sequence of sequences $\left((X_i^{(n)})_{1 \leq i \leq n}\right)_{n \in \mathbb{N}}$. For a given n, the $X_i^{(n)}$ are identically distributed Gaussian random variables.*

*Let $0 \leq w, x, y, z \leq 1$ and $r(w, x, y, z) := \frac{\rho(|z-x|)-\rho(|z-w|)-\rho(|y-x|)+\rho(|y-w|)}{2\sqrt{1-\rho(|x-w|)}\sqrt{1-\rho(|z-y|)}}$. Then:*

$$\lim_{n\to\infty} \mathbb{V}(\tau_n) = \frac{16}{\pi} \int_0^1 (1-z) \int_0^z \int_0^y f(x, y, z) \mathrm{d}x\mathrm{d}y\mathrm{d}z, \tag{9}$$

*where $\tau_n$ is the Mann-Kendall tau of the $(X_i^{(n)})_{1 \leq i \leq n}$ sequence and $f(x, y, z) = \arcsin(r(0, x, y, z)) + \arcsin(r(0, y, x, z)) + \arcsin(r(0, z, x, y))$.*

In the field of time series analysis, stationary autoregressive moving average (ARMA) processes are often considered. They are composed of an autoregressive part and a moving average part. Since these latter processes are the examples we are interested in, we introduce the more general class of ARMA processes.



**Definition 1** (ARMA process). *An autoregressive moving average process of order (p,q) (ARMA(p,q)) is a discrete temporal process ($X_i, i \in \mathbb{N}$) such that:*

$$X_i = \epsilon_i + \sum_{j=1}^{p} k_j X_{i-j} + \sum_{j=1}^{q} \theta_j \epsilon_{i-j} \qquad (10)$$

*where $k_j$ and $\theta_j$ are the parameters of the model and the $\epsilon_j$ are the error terms (white noise).*

*An autoregressive process AR(p) is an ARMA(p,0).*

*A moving average MA(q) is an ARMA(0,q).*

*In the following, we will only consider $p = 1$, $0 < k < 1$, $\epsilon_j \sim \mathcal{N}(0, 1 - k^2)$ i.i.d. Gaussian random variables and all the $\theta_j$ equal to one (the simple moving average, SMA).*

In the definition of the ARMA process, the noise $\epsilon_j$ is chosen as Gaussian and so the ARMA process is a Gaussian process. Moreover, we choose the variance of $\epsilon_j$ such that for the AR(1) process, $X_i \sim \mathcal{N}(0, 1)$.

**Lemma 2** (Correlation for the ARMA process). *Let ($X_i, i \in \mathbb{N}$) follow an ARMA(1,q−1) process such that:*

$$X_i = \epsilon_i + k X_{i-1} + \sum_{j=1}^{q-1} \epsilon_{i-j}, \quad \epsilon_j \overset{iid}{\sim} \mathcal{N}(0, 1 - k^2). \qquad (11)$$

*Let $0 \leq i \leq j$ and $d = j - i$, then:*

• *If $d < q-1$*

$$\mathrm{corr}(X_i, X_j) = \frac{(q - d)(1 - k^2) + k(k^{q+d} + k^{q-d} - 2k^d)}{q(1 - k^2) - 2k(1 - k^q)}. \qquad (12)$$

• *If $d \geq q - 1$*

$$\mathrm{corr}(X_i, X_j) = \frac{(1 - k^q)^2 k^{d+1-q}}{q(1 - k^2) - 2k(1 - k^q)}. \qquad (13)$$

Let $0 < k < 1$, $a > 0$, $n \in \mathbb{N}$, and let $(X_i^{(n)})_{1 \leq i \leq n}$ follow an ARMA(1,$q_n$) process of parameter $k_n$ with $(k_n)^{n-1} \xrightarrow[n \to \infty]{} k_{\mathrm{tot}}$ and $q_n/n \xrightarrow[n \to \infty]{} a$. Then Assumption 2 holds, and we can use Lemma 1 and Lemma 2 to obtain the following theorem:

**Theorem 1** (Renormalized correlation function for the ARMA process). *Let $a > 0$ and $0 < k_{\mathrm{tot}} < 1$. For $n \geq 1$, let $(X_i^{(n)})_{1 \leq i \leq n}$ follow an ARMA(1,$q_n$) process of parameter $k_n$ and such that $q_n/n \xrightarrow[n \to \infty]{} a$, $(k_n)^{n-1} \xrightarrow[n \to \infty]{} k_{\mathrm{tot}}$, then:*

$$\rho(x) = \begin{cases} \dfrac{(a-x)\log(k_{\mathrm{tot}}) + (k_{\mathrm{tot}}^x - k_{\mathrm{tot}}^{a+x}/2 - k_{\mathrm{tot}}^{a-x}/2)}{a\log(k_{\mathrm{tot}}) + (1 - k_{\mathrm{tot}}^a)} & \text{if } 0 \leq x \leq \min(a, 1) \\[2ex] \dfrac{(1 - k_{\mathrm{tot}}^a)^2 k_{\mathrm{tot}}^{x-a}}{-2(a\log(k_{\mathrm{tot}}) + (1 - k_{\mathrm{tot}}^a))} & \text{if } a \leq x \leq 1 \end{cases}$$

From Theorem 1, we deduce results for the AR(1) and SMA(q) cases.

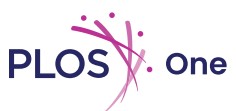

**Corollary 1.1** (Asymptotic variance for the AR(1) process). *Let* $(X_i^{(n)})_{1 \leq i \leq n}$ *be a sequence of random variables following an* AR*(1) process of parameter* $k_n$ *such that* $(k_n)^{n-1} \xrightarrow[n \to \infty]{} k_{\text{tot}}$ *with* $0 < k_{\text{tot}} < 1$. *Then,* $\rho(x) = k_{\text{tot}}^x$ *and:*

$$\lim_{n \to \infty} \mathbb{V}(\tau_n) \geq \frac{32}{\pi} \int_0^1 (1-z) \int_0^z \int_0^y \arcsin\left( \frac{\sqrt{1 - k_{\text{tot}}^{y-x}}\left(k_{\text{tot}}^{z-y}(1 - \sqrt{1 - k_{\text{tot}}^x}\sqrt{1 - k_{\text{tot}}^z}) + k_{\text{tot}}^x\right)}{4\sqrt{1 - k_{\text{tot}}^z}} \right) dx\,dy\,dz > 0.$$

**Corollary 1.2** (Asymptotic variance for the SMA(q) process). *Let* $(X_j^{(n)})_{1 \leq j \leq n}$ *be a sequence of random variables following a* SMA*(*$q_n$*) process with* $q_n/n \xrightarrow[n \to \infty]{} a, a > 0$. *Then,* $\rho(x) = \max(1 - \frac{x}{a}, 0)$.

- *If* $a \geq 1$ :

$$\lim_{n \to \infty} \mathbb{V}(\tau_n) = \frac{17}{72}.$$

- *If* $0 < a < 1$ :

$$\lim_{n \to \infty} \mathbb{V}(\tau_n) \geq \frac{17}{72}a^3(4 - 3a).$$

In particular, in Corollaries 1.1 and 1.2, $\frac{\tau_n}{\sqrt{\mathbb{V}(\tau_n)}}$ cannot converge in distribution to a Gaussian. Indeed, let's assume that $\lim_{n \to \infty} \mathbb{V}(\tau_n) = \ell > 0$. Then, as $\tau_n$ is bounded between $-1$ and $+1$, the support of the distribution of $\frac{\tau_n}{\sqrt{\mathbb{V}(\tau_n)}}$ is uniformly bounded for all $n \in \mathbb{N}$ and so the normalized Mann-Kendall tau cannot converge to a Gaussian.

A key result regarding a specific sum of arcsin terms, which is essential for calculating the variance of the Mann-Kendall tau in the context of SMA processes, is presented in the following Proposition:

**Proposition 1.**

$$\forall n \geq 3, \quad \frac{1}{\binom{n+1}{4}} \sum_{0 \leq i < j < k < l \leq n} \arcsin\left( \frac{k-j}{\sqrt{k-i}\sqrt{l-j}} \right) = \frac{\pi}{6}.$$

In this section, we demonstrated that the asymptotic (as $n \to \infty$) variance of the Mann-Kendall tau is strictly positive for sequences of time series of length $n$ generated by the following processes:

- an AR(1) process with autocorrelation at lag-1 parameter $k_n$ increasing towards 1 as $n \to \infty$ in the following manner: $k_n = k_{\text{tot}}^{1/(n-1)}$ with $k_{\text{tot}} \in ]0, 1[$ independent of $n$.
- an SMA($q_n$) process with parameter $q_n = \lfloor an \rfloor$, which corresponds to an asymptotic relative window size $\alpha = \frac{a}{a+1}$.

Consequently, the normalized Mann-Kendall tau $\frac{\tau_n}{\sqrt{\mathbb{V}(\tau_n)}}$ of these time series cannot converge to a Gaussian.

## 4 Checking for non-normality

In practical applications, one deals with finite time series produced by AR(1) or SMA(q) processes with usually constant parameter values $k$ or $q$. In that case, we have seen at the end of Sect 2 that the asymptotic distribution of the Mann-Kendall tau is Gaussian. However, this asymptotic result is less relevant when it is needed to decide whether a statistic

calculated on a finite time series would approximately follow a Gaussian distribution or not. It is possible to determine the parameters $k_{tot}$ or $\alpha$ so that the time series would have been produced by processes described in Examples 1 and 2. For such processes we know that these parameters determine the asymptotic variance associated with increasing upsampling. For example, a time series of length $n$ produced by an AR(1) of correlation at lag-1 $k$ can be identified as a time series of length $n$ described by Example 1, where $k_{tot} = k^{n-1}$. If $k_{tot}$ is asymptotically (as $n \to \infty$) non-zero, we have proved that the distribution of the Mann-Kendall tau of this time series cannot converge to a Gaussian distribution. Consequently, our results suggest that, for time series of finite length, assuming normality of the Mann-Kendall statistic is not appropriate when autocorrelation is strong relative to the length of the time series, and this also limits the applicability of the modified Mann-Kendall tests. Therefore, it is essential to understand how the distribution of the Mann-Kendall tau deviates from normality for specific parameter values and time series lengths.

Deriving Berry-Esseen bounds for the Mann-Kendall tau across different types of autocorrelated processes would help us understanding how the distribution of the Mann-Kendall tau deviates from normality for specific parameter values and time series lengths, as these bounds quantify the accuracy of the Gaussian approximation [17]. For instance, uniform and non-uniform bounds have been established for *U-statistics* with symmetric kernels in the case of independent samples [18] and weakly dependent samples [19]. However, these bounds depend on both sample size $n$ and a constant term influenced by autocorrelation, which lacks a clear expression. It makes it impractical as a theoretical criterion for when to use the family of Mann-Kendall tests for autocorrelated processes such as the AR(1) process. Furthermore, while the Kendall tau is a *U-statistic* with a symmetric kernel, the Mann-Kendall tau has a non-symmetric kernel [16], limiting the applicability of many classical results. Therefore, we conducted a numerical investigation on the Mann-Kendall tau distribution for time series of length $n$ generated by AR(1) and SMA(q) processes, with varied levels of autocorrelation (parameter $k$) and window size (parameter $q$). We investigated whether isolines of the values of parameters $k_{tot}$ or $\alpha$ on the spaces delineate regions where the distribution of the Mann-Kendall tau is close to Gaussian or not.

### 4.1 Numerical investigation

For each combination of parameters and length of time series, we computed the Mann-Kendall tau of $10^2$ different time series to find the empirical distribution of tau. To evaluate if these empirical distributions are roughly Gaussian, we used the Shapiro-Wilk test, which tests the null hypothesis that the population is an i.i.d. Gaussian sample (with unknown expectation and variance) [20]. We compared the rejection rate (computed over $10^4$ $p$-values) of this null hypothesis to the predetermined significance level. If the true distribution of our simulated tau-values is, indeed, Gaussian, then the proportion of rejections should converge to the significance level as sample size (i.e., the number of tau-values) increases. We used the Shapiro-Wilk test because it is more powerful than other classic normality tests [21], utilizing the stats.shapiro Python implementation from the SciPy package [22].

### 4.2 For the AR(1) process

First of all, we studied time series of length $n$ produced by AR(1) processes with autocorrelation at lag-1 $k$, for values of $n$ ranging from 5 to 100 and $k$ from 0.23 to 0.99. Fig 1A shows the proportion at which the Shapiro-Wilk test rejects normality at the 5% significance level depending on $n$ and $k$.

It can be seen that, for a fixed time series of length $n$, the distribution of the Mann-Kendall tau is not Gaussian if the autocorrelation parameter $k$ of the generating AR(1) is too close to 1. Furthermore, the closer $k$ is to 1, the larger $n$ needs to be for the distribution of the Mann-Kendall tau to remain (approximately) Gaussian. This suggests that, for each $k$, there exists a minimum $n$ above which the distribution of Mann-Kendall tau can be considered Gaussian.

As proved in Sect 3, the distribution of the Mann-Kendall tau of time series where $k_{tot} = k^{n-1}$ is asymptotically non-zero cannot converge to a Gaussian. Lines where $k_{tot}$ is constant are added in black on Fig 1A for four examples of $k_{tot}$ values. Furthermore, the ranges of values for discriminating empirical rejection rates have been chosen to match the proportion of



**Fig 1. Deviations from the Gaussian distribution for AR(1) processes.** Empirical proportion of rejections of the null hypothesis of the Shapiro-Wilk test of normality at the 5% significance level for AR(1) processes. $p$-values are calculated over $10^2$ values of the Mann-Kendall statistic, and rejection proportions are based on $10^4$ $p$-values. A: Rejection rate of time series of length $n \in \{5, 10, \ldots, 100\}$ generated by AR(1) processes of autocorrelation at lag-1 $k \in [0.23, 0.99]$. Examples of lines for which $k_{tot} = k^{n-1}$ is constant are in black, for four values of $k_{tot}$. B: Rejection rate of time series of length $n$ generated by AR(1) processes for several values of $k_{tot}$. The dotted line is the 5% significance threshold. If the true distribution of the Mann-Kendall tau is Gaussian for a given $k_{tot}$, then the proportion of rejection should converge to the significance level as sample size (i.e., the number of tau-values) increases. Fig C-H present examples of empirical distributions of the normalized Mann-Kendall tau $\frac{\tau}{\mathbb{V}(\tau)}$ for $n = 50$ and several values of $k_{tot}$.

rejection associated with each $k_{tot}$. Then, we see that the lines where $k_{tot}$ is constant are also the lines where the proportion of rejection of the Shapiro-Wilk test is constant. Thus, $k_{tot} = k^{n-1}$ is the right scaling to decide in practice whether the Gaussian approximation is justified.

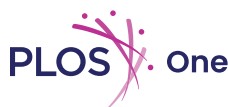

Moreover, the closer $k_{tot}$ is to 1 ($k_{tot} \in ]0, 1[$), the further the normality test rejection rate is from the significance level. This is the expected effect: larger autocorrelation drives the Mann-Kendall tau distribution away from the Gaussian distribution. However, if $k_{tot}$ is small enough, the proportion of rejection is approximately the significance level. This confirms that for small $k_{tot}$, the distribution of the Mann-Kendall tau is well approximated by a Gaussian. Fig 1B presents rejection rates of the null hypothesis of the Shapiro-Wilk test depending on $n$, for several values of $k_{tot}$. It is clear that the rejection rates are approximately constant for a given value of $k_{tot}$, for any $n$. We also see that the rejection rates are very close to their final values for small values of $n$ (typically for $n > 10$), making these results useful for short time series.

This numerical study validates the theoretical scaling obtained in Sect 3, but also provides practical values of $k_{tot}$ for which the Gaussian approximation is not adequate. As the Shapiro-Wilk test is slightly conservative, rejection rates converge to values which are slightly below 5% when $k_{tot}$ goes to 0. Here, we see on Fig 1A and 1B that the proportion of rejection is equal to the significant threshold for $k_{tot} \approx 10^{-8}$. Then, we propose to chose this as a criterion to decide whether the Gaussian approximation is justified. Examples of empirical distributions of the Mann-Kendall tau for several $k_{tot}$ values are shown in Fig 1C–1H. We see that for $k_{tot} > 10^{-8}$, the Gaussian approximation does not seem justified. If $k_{tot}$ is close enough to 1, the empirical distribution is bimodal. We note that the distribution for intermediate values of $k_{tot}$ (see Fig 1E for example) is very similar to the one found by Hamed [7], who proposed the Beta distribution as a more accurate approximation.

Anyone who wants to use a modified Mann-Kendall test for autocorrelated data on a time series from a real system, and assumes that the underlying process is an AR(1) process with autocorrelation at lag-1 $k$ can therefore estimate the total autocorrelation parameter $k_{tot} = k^{n-1}$ and know if the Gaussian approximation is justified. This allows to decide whether modified Mann-Kendall test can be applied or not on the Mann-Kendall tau of the time series. For example, let's consider a time series of length $n = 20$ produced by an AR(1) process with autocorrelation at lag-1 parameter $k = 0.5$. Then, $k_{tot} = k^{n-1} = 0.5^{19} \approx 2 \times 10^{-6}$. According to the previous criterion, $2 \times 10^{-6} > 10^{-8}$ is clearly too high to consider that the distribution of the Mann-Kendall tau of the time series of interest is Gaussian at the 5% significance level. Therefore, it is not reasonable to assume that the distribution of the Mann-Kendall tau is Gaussian and therefore to apply a test from the family of modified Mann-Kendall tests for autocorrelated data as they rely on this Gaussian assumption. In this case, these tests are not suitable for reliably detecting trends.

**4.2.1 Case study.** We apply the practical criterion presented in the last section - that is to compare the value of $k_{tot}$ to $10^{-8}$ - to time series from the literature. In particular, we focus on the three papers that have proposed modified Mann-Kendall tests for autocorrelated data [3,4,7]. The analyzed time series are taken from the hydrological literature.

The existence of a trend in a time series alters the estimate of the autocorrelation parameter [4]. Therefore, we follow the method proposed by Yue and Wang [4] and first remove any potential trend, using the non-parametric method of Theil [23] and Sen [24]. Then, the lag-1 autocorrelation coefficient $\hat{k}$ is estimated on the detrended time series $(X_i)_{1 \le i \le n}$ [25]:

$$\hat{k} = \frac{\frac{1}{n-1} \sum_{i=1}^{n-1} (X_i - \bar{X})(X_{i+1} - \bar{X})}{\frac{1}{n} \sum_{i=1}^{n} (X_i - \bar{X})^2}, \tag{14}$$

with $\bar{X}$ the sample average.

As the results from Sect 3 concern AR(1) processes with positive autocorrelation parameters, we only keep time series which satisfy this condition. The length of the time series $n$, the estimated autocorrelation at lag-1 parameter $\hat{k}$, the upper bound of the 90% confidence interval (upper bound of the confidence interval for a one-sided test at the significance threshold of 5%) $u(\hat{k})$, as well as the estimated total autocorrelation $u(\hat{k}_{tot})$ of the upper bound, and the identifiers of the stations where the measurements were taken (see the cited articles for the full identification of the time series) are shown in Table 1. See the data availability section for the python implementation.

**Table 1**. **Validity of the Gaussian approximation for example time series.**

| Article | Station ID | River name | n | $\hat{k}$ | $u(\hat{k})$ | $u(\hat{k}_{\text{tot}})$ | Gaussian approximation |
|---|---|---|---|---|---|---|---|
| [3] | 05464500 | Cedar River | 90 | 0.30 | 0.47 | $1.0 \times 10^{-29}$ | ✓ |
| [4] | 08CE001 | Stikine river | 32 | 0.10 | 0.39 | $1.5 \times 10^{-13}$ | ✓ |
| | 08DA005 | Surprise creek | 28 | 0.23 | 0.54 | $5.7 \times 10^{-8}$ | - |
| | 09AA006 | Atlin river | 45 | 0.25 | 0.49 | $2.6 \times 10^{-14}$ | ✓ |
| | 09AA015 | Wann river | 29 | 0.28 | 0.59 | $3.2 \times 10^{-7}$ | - |
| | 10EB001 | South nahanni river | 25 | 0.03 | 0.36 | $2.7 \times 10^{-11}$ | ✓ |
| | 02YA001 | St. genevieve river | 27 | 0.12 | 0.43 | $4.0 \times 10^{-10}$ | ✓ |
| | 02VC001 | Romaine (riviere) | 37 | 0.15 | 0.42 | $2.6 \times 10^{-14}$ | ✓ |
| | 06CD002 | Churchill river | 33 | 0.53 | 0.81 | $1.2 \times 10^{-3}$ | - |
| | 08HA003 | Koksilah river | 37 | 0.16 | 0.43 | $4.8 \times 10^{-14}$ | ✓ |
| [7] | 1134100 | Niger | 12 | 0.25 | 0.72 | $2.8 \times 10^{-2}$ | - |
| | 4214210 | Beaver | 16 | 0.28 | 0.69 | $3.8 \times 10^{-3}$ | - |
| | 6335301 | Main River | 15 | 0.08 | 0.50 | $6.7 \times 10^{-5}$ | - |
| | 6335500 | Main | 12 | 0.01 | 0.49 | $3.6 \times 10^{-4}$ | - |

Estimated autocorrelation parameter at lag-1 $\hat{k}$ of the detrended time series, its upper bound $u(\hat{k})$, estimated total autocorrelation $u(\hat{k}_{\text{tot}})$ of the upper bound, length of the time series $n$, Station ID and river name to identify concerned time series in cited articles. Last column indicates if the Gaussian approximation is appropriate (✓) according to the practical criterion. The data are taken from [3], [4] and [7].

Comparing $u(\hat{k}_{\text{tot}})$ with the $10^{-8}$ threshold, we conclude that the Gaussian approximation for the Mann-Kendall tau is suitable for seven out of the fourteen time series. The time series concerned are indicated by a tick in the last column of Table 1. For the remaining seven time series, this implies that a portion of the $\hat{k}_{\text{tot}}$ 95% confidence interval does not satisfy the previously established criterion.

The length of the time series $n$, the estimated autocorrelation at lag-1 parameter $\hat{k}$, the upper bound of the 90% confidence interval (upper bound of the confidence interval for a one-sided test at the significance threshold of 5%) $u(\hat{k})$, as well as the estimated total autocorrelation $u(\hat{k}_{\text{tot}})$ of the upper bound

### 4.3 For the SMA process

We applied the same methodology to study time series of length $n$ produced by SMA processes of order $q$, for values of $n$ ranging from 5 to 150 and $q$ from 5 to 50. Fig 2A shows the proportion of time series for which the Shapiro-Wilk test rejects normality at the 5% significance level as a function of $n$ and $q$.

It can be seen that, for a time series of fixed length $n$, the distribution of the Mann-Kendall statistic is not Gaussian if the order $q$ of the generating SMA process is too high. Furthermore, the higher $q$ is, and the larger $n$ needs to be, for the distribution of the Mann-Kendall tau to remain (approximately) Gaussian. This suggests that, for each $q$, there exists a minimum $n$ above which the distribution of tau can be considered Gaussian.

As proved in Sect 3, the distribution of the Mann-Kendall tau of time series where the relative window size $\alpha = \frac{q}{N} = \frac{q}{n+q-1}$ is asymptotically non-zero cannot converge to a Gaussian (see Example 2 for notations). Lines for which $\alpha$ is constant are added in black on Fig 2A for four $\alpha$ values.

Furthermore, the ranges of rejection rates for each colour band have been chosen to minimize the distances from the values of $\alpha$ plotted. Then, we see that the lines where $\alpha$ is constant are also the lines where the proportion of rejection of the Shapiro-Wilk test is constant. Thus, $\alpha$ is the right scaling to decide in practice whether the Gaussian approximation is justified.

Moreover, the closer $\alpha$ is to 1 ($\alpha \in [0, 1]$), the further the normality test rejection rate is from the significance level. This is the expected effect: larger window sizes introduce more autocorrelation, thus driving the Mann-Kendall tau distribution away from the Gaussian distribution. However, if $\alpha$ is small enough, the proportion of rejection is approximately

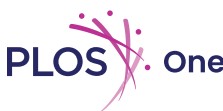

**Fig 2. Deviations from the Gaussian distribution for SMA processes.** A: Empirical proportion of rejections of the null hypothesis of the Shapiro-Wilk test of normality at the 5% significance level for time series of length $n$ and initial length $N$ (see Example 2) generated by SMA processes of order $q$. Examples of lines for which the relative window size $\alpha = \frac{q}{N} = \frac{q}{n+q-1}$ is constant are in black, for four examples of $\alpha$. B: Rejection rate of time series of initial length $N$ generated by SMA processes for several values of $\alpha$. The dotted line is the 5% significance threshold. If the true distribution of the Mann-Kendall tau is Gaussian for a given $\alpha$, then the proportion of rejection should converge to the significance level as sample size (i.e., the number of tau-values) increases. Fig C–F present examples of empirical distributions of the normalized Mann-Kendall tau $\frac{\tau}{\mathbb{V}(\tau)}$ for several values of $\alpha$ when $N = 100$.

the significance level. This confirms that for small $\alpha$, the distribution of the Mann-Kendall tau is well approximated by a Gaussian.

Fig 2B presents rejection rates of the null hypothesis of the Shapiro-Wilk test depending on $N$, for several values of $\alpha$. It is clear that the rejection rates are approximately constant for all values of $N$ if $\alpha$ is fixed. Rejection rates would similarly be constant for all values of $n$ if $\alpha$ is fixed. We also see that the rejection rates are very close to their final values for small values of $N$ (typically for $N > 10$), making these results useful for short time series.

Therefore, this numerical study validates the theoretical scaling obtained in Sect 3, but also provides practical values of $\alpha$ for which the Gaussian approximation is not adequate. As the Shapiro-Wilk test is slightly conservative, rejection rates converge to values slightly below 5% (which would be the expected rate for independent data) when $\alpha$ goes to 0. Here, we see on Fig 2A and 2B that the proportion of rejection is equal to the significant threshold for $\alpha \approx 10\%$. Then, we propose to use this as a criterion to decide whether the Gaussian approximation is justified. Note that, for a relative window size of 10% or less, the Shapiro-Wilk test does not reject normality, but it does not prove either that the distribution is normal (see Discussion).

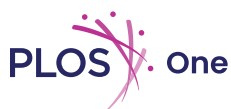

Examples of empirical distributions of the Mann-Kendall tau for several $\alpha$ values are shown in Fig 2C–2F. We see that for $\alpha > 0.10$, the Gaussian approximation does not seem justified visually. If $\alpha$ is high enough, the empirical distribution is bimodal, very far from the Gaussian distribution. Distributions of the Mann-Kendall statistic are very similar to the one found for the autoregressive process, see Fig 1.

Anyone who wants to use a modified Mann-Kendall test on data produced by the averaging on windows of size $q$ of an initial time series of length $N$ from a real system can therefore estimate the relative window size $\alpha = \frac{q}{N}$ and know if the Gaussian approximation is justified. This allows to decide whether modified Mann-Kendall test can be applied or not on the Mann-Kendall tau of the time series. For example, if considering the averaging of a time series of length $N = 50$ over rolling windows of size $q = 20$, then, the relative window size $\alpha = \frac{q}{N} = \frac{20}{50} = 0.4$ and according to the previous criterion, $\alpha = 0.4$ is too high to consider that the distribution of the Mann-Kendall tau of the time series of interest is Gaussian. Therefore, it is not reasonable to apply a test from the family of modified Mann-Kendall tests for autocorrelated data to reliably detect trends.

From the previous section, we conclude that the distribution of the Mann-Kendall tau for time series generated by SMA processes is not approximately Gaussian for relative window sizes of more than 10%, independently of the time series length.

## Discussion

In this article, we demonstrate the existence of sequences of time series generated by stationary autoregressive AR(1) and simple moving average (SMA) processes for which the normalized Mann-Kendall tau distribution cannot be asymptotically Gaussian. Instead, it converges to a bounded distribution with strictly positive variance. This result suggests that the non-Gaussian nature of the distribution should emerge noticeably in finite-length time series with sufficient autocorrelation. We found in a numerical investigation that the parameters determining the variance in our asymptotic results are indeed the ones which determine whether the distribution of the Mann Kendall tau will be close to Gaussian or not in finite-length time series. To guide practical application, we provided easy-to-implement criteria which clarify when the Gaussian approximation is appropriate for tests applied to real data. Our numerical investigations indicate that these criteria remain relevant when applied to time series of a relatively small number of points.

For time series of length $n$ generated by an AR(1) process with lag-1 parameter $k$, we showed theoretically that $k_{\text{tot}} = k^{n-1}$ emerges naturally as the right scaling between $k$ and $n$ to reject the asymptotic Gaussian approximation. Based on these theoretical foundations, we selected the Shapiro-Wilk test for normality to numerically check that $k_{\text{tot}}$ is the correct scaling. We also proposed the practical threshold $k_{\text{tot}} = 10^{-8}$ to decide whether the distribution of the Mann-Kendall tau of time series is Gaussian or not. It corresponds to the $(k,n)$ contour where the null hypothesis of a Gaussian distribution is rejected in 5% of tests. The criteria on autocorrelation of a timeseries can be checked using the code shared in the data accessibility Section.

Regarding the SMA process of order $q$, the correct scaling is $\alpha = \frac{q}{q+n-1} = \frac{q}{N}$ which can naturally be interpreted as the relative window size of the moving average. Numerically, using the Shapiro-Wilk test for normality, we find that if $\alpha$ is larger than 10%, then a Gaussian distribution of the Mann-Kendall tau is rejected in over 5% of the tests. Therefore, we propose $\alpha = 10\%$ as a practical threshold to decide whether the distribution of the Mann-Kendall tau of time series is Gaussian or not. For real datasets, if one assumes that the data are resampled using a moving average but does not know the parameter $\alpha$, one could investigate the shape of the autocorrelation function. Theoretically, its slope could be used to retrieve the value of the relative window size. However, further analysis is required to evaluate the power of this approach to detect the true value of the relative window size.

We illustrated these results with empirical distributions of the normalized Mann-Kendall tau for several values of $k_{\text{tot}}$ and $\alpha$. The distributions are very similar for all $n$, depending mainly on the value of $k_{\text{tot}}$ and $\alpha$, which once again underlines the fact that the proposed scalings can be used to determine whether a distribution is Gaussian or not in finite-length time

 

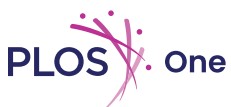

series. These findings fill a gap between intermediate-length observations, as discussed by [7], and different asymptotic results, which predict a Gaussian distribution in cases where autocorrelation is present in the data but $k_{\text{tot}}$ converges to zero in long time series.

Nevertheless, our results should be treated with some caution. We simulated the stochastic processes assuming error terms and initial observations sampled from independent Gaussian variables. This does not need to be the case for observed time series. We have not studied the degree to which the various Mann-Kendall tests might be robust with respect to such deviations from the normality assumptions on error terms and initial observations. Future work could investigate whether our results hold when relaxing the assumption that the underlying process is Gaussian, which was necessary for Eq (4). Additionally, deriving Berry-Esseen bounds for the Mann-Kendall tau across different types of auto-correlated processes would enhance understanding, as these bounds quantify the accuracy of the Gaussian approximation. In time series where $k_{\text{tot}} > 10^{-8}$ in the case of AR(1) processes and where $\alpha > 0.10$ for SMA processes, however, we don't immediately see how such deviations might suddenly generate statistics which follow a Gaussian distribution. In time series where $k_{\text{tot}} < 10^{-8}$ and where $\alpha < 0.10$, this might lead to additional cases where statistics are not Gaussian. Furthermore, the sample size for the Shapiro-Wilk test was arbitrarily set to $10^2$ time series. Changing the sample size could slightly change the rejection rates of the null hypothesis; however, an exploratory analysis showed that this has minimal impact on our results and the identified thresholds.

Although Lemma 1 is broadly applicable, our focus has been on ARMA processes due to their widespread use in scientific fields such as ecology [26], hydrology [25], and finance [27]. Autoregressive models, particularly the AR (1) process, are fundamental for time series analysis, modeling, and forecasting, with frequent applications in hydrology [4]. In contrast, moving-average models are often used for noise reduction in time series, as well as for modeling purposes. Among different statistical techniques for forecasting and trend detection involving regression methods, time series methods and stochastic processes (e.g., [28–30]), the Mann-Kendall statistic can be used to test the null hypothesis of no trend without fitting a specific time series or non-linear model. Mann-Kendall statistics with Gaussian distributions were recently recommended to this end [31]. We believe that our findings could encourage preliminarily testing the normality assumption before applying Mann-Kendall tests for trend detection. Consequently, our results on the importance of autocorrelations may have broader relevance, particularly when testing for the presence of critical transitions in time series (also known as early warning signals), where methodology involves averaging, which may introduce artificial autocorrelation. We plan to explore these implications further in a forthcoming paper.

## Supporting information

**S1 Appendix. Proofs of the main text.**
(PDF)

## Acknowledgments

The authors are very grateful to Michael Kopp and Jean-René Chazottes for helpful discussions and for providing useful comments on various versions of this manuscript.

## Author contributions

**Conceptualization:** Tristan Gamot, Nils Thibeau–Sutre, Tom J. M. Van Dooren.

**Formal analysis:** Tristan Gamot, Nils Thibeau–Sutre.

**Software:** Tristan Gamot, Nils Thibeau–Sutre.

**Validation:** Tristan Gamot, Nils Thibeau–Sutre, Tom J. M. Van Dooren.



**Writing – original draft:** Tristan Gamot, Nils Thibeau–Sutre.

**Writing – review & editing:** Tristan Gamot, Nils Thibeau–Sutre, Tom J. M. Van Dooren.

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
