## [Decision Letter · Decision Letter 0]

26 Nov 2025

PONE-D-25-48903

On the Gaussian distribution of the Mann-Kendall tau in the case of autocorrelated data

PLOS ONE

Dear Dr. Thibeau--Sutre,

Thank you for submitting your manuscript to PLOS ONE. After careful consideration, we feel that it has merit but does not fully meet PLOS ONE’s publication criteria as it currently stands. Therefore, we invite you to submit a revised version of the manuscript that addresses the points raised during the review process.

We look forward to receiving your revised manuscript.

Kind regards,

Sandip V George, PhD

Academic Editor

PLOS ONE

Journal Requirements:

[AAP2023 FIRE mini-grant].

4. We are unable to open your Supporting Information/other file [Code_Figures.ipynb]. Please kindly revise as necessary and re-upload.

Additional Editor Comments:

Thank you for your submission. The work was reviewed by two reviewers, both of whom suggested some revisions which would make the manuscript stronger. In particularly, I would recommend addressing the suggestions of the second reviewer.

Reviewers' comments:

Reviewer's Responses to Questions

**Comments to the Author**

1. Is the manuscript technically sound, and do the data support the conclusions?

Reviewer #1: Yes

Reviewer #2: Yes

2. Has the statistical analysis been performed appropriately and rigorously?

Reviewer #1: Yes

Reviewer #2: Yes

3. Have the authors made all data underlying the findings in their manuscript fully available?

Reviewer #1: Yes

Reviewer #2: Yes

4. Is the manuscript presented in an intelligible fashion and written in standard English?

Reviewer #1: Yes

Reviewer #2: Yes

5. Review Comments to the Author

Reviewer #1: The manuscript explores an asymptotic framework for autoregressive processes of order 1 and simple moving 12average processes of order q. The authors prove that, along with upsampling sequences, the distribution of the normalized Mann-Kendall tau does not converge to a Gaussian.

On the whole, the subject of this manuscript is interesting. The statistical model and the equations are derived well, and the numerical results are adequate. However, the manuscript needs minor corrections which are listed in the attached file.

Reviewer #2: I commend the authors for choosing a vital theme in longitudinal analyses: that of modifying asymptotic guarantees to finite-sample contexts and I like their organization, analysis and presentation. Some comments for the authors to consider:

Is it possible to upgrade one step higher on your first example: from an autoregressive to a threshold autoregressive (TAR) process? In particular, if the finiteness doesn’t impact the pre-threshold phase and the post-threshold phase equally? How would the scaling alter? Maybe some simulations here?

I’d encourage the authors to enlarge the discussions a bit more. For instance, I’d be curious to know the impacts these may have on detecting changes in trend and how would those compare with other recent techniques such as e-divergence or prophet-type strategies. A mention or acknowledgement of these would go a long way even if the authors do not deploy a large-scale analysis to examine these things on the current paper.

Maybe motivate the analyses (or mention in the discussions) with some non-time series anecdotes/contexts where the departure (from theoretical asymptotics) due to finiteness had a crucial role to play in policy, people’s lives, etc.

I like the approach the authors have taken, especially in defining/stressing repeatedly what they are trying to achieve (point out a deviation from normality), but we can be mindful of Plos One’s diverse readership. If in a small fresh section, the authors can add small numerical examples to clarify structures that may seem technical to some (sequence of sequences, etc.), that could ensure wider accessibility with the article being self-contained (i.e., without the readers having to visit references (hydrological, for instance) the authors have cited). If this exceeds length requirements, please ignore.

Do the authors plan on developing software packages (for example, an R package) where someone can enter a finite-length time series and out will come a decision (through their Shapiro-Wilks choice) that says whether normality is okay and/or output the right scaling in case it is not? They may mention this as a future possibility as well.

Some minor concerns that another round of edits could allay:

For example, on such lines as 34-36, page2: “Concerning the SMA(q) process, these time series amount to 34

averaging a larger and larger sample of a white noise process with a constant relative 35

window size, which can be used as the null model when testing for the presence of 36

critical transitions in time series - also known as early warning signals [8].”

- “Regarding” instead of “concerning”? Did you mean a “larger and larger sample” or a “larger sample”. If the former, i.e., a sequential change-point-type approach, please consider being more explicit.

6. PLOS authors have the option to publish the peer review history of their article (what does this mean?). If published, this will include your full peer review and any attached files.

Reviewer #1: **Yes:** Younus Al-Taweel

Reviewer #2: **Yes:** Moinak Bhaduri

---

## [Author Response · Author response to Decision Letter 1]

13 Jan 2026

Dear editor,

As requested, we uploaded the "Response to Reviewers" file in the Attach Files section. It contains all the requested responses.

Best regards,

The authors

---

## [Decision Letter · Decision Letter 1]

26 Jan 2026

On the Gaussian distribution of the Mann-Kendall tau in the case of autocorrelated data

PONE-D-25-48903R1

Dear Dr. Thibeau--Sutre,

We’re pleased to inform you that your manuscript has been judged scientifically suitable for publication and will be formally accepted for publication once it meets all outstanding technical requirements.

Kind regards,

Sheng Du

Academic Editor

PLOS One

Additional Editor Comments (optional):

Reviewers' comments:

Reviewer's Responses to Questions

**Comments to the Author**

1. If the authors have adequately addressed your comments raised in a previous round of review and you feel that this manuscript is now acceptable for publication, you may indicate that here to bypass the “Comments to the Author” section, enter your conflict of interest statement in the “Confidential to Editor” section, and submit your "Accept" recommendation.

Reviewer #1: All comments have been addressed

Reviewer #2: All comments have been addressed

2. Is the manuscript technically sound, and do the data support the conclusions?

Reviewer #1: Yes

Reviewer #2: Yes

3. Has the statistical analysis been performed appropriately and rigorously?

Reviewer #1: Yes

Reviewer #2: Yes

4. Have the authors made all data underlying the findings in their manuscript fully available?

Reviewer #1: Yes

Reviewer #2: Yes

5. Is the manuscript presented in an intelligible fashion and written in standard English?

Reviewer #1: Yes

Reviewer #2: Yes

6. Review Comments to the Author

Reviewer #1: (No Response)

Reviewer #2: (No Response)

7. PLOS authors have the option to publish the peer review history of their article (what does this mean?). If published, this will include your full peer review and any attached files.

Reviewer #1: **Yes:** Younus Al-Taweel

Reviewer #2: **Yes:** Moinak Bhaduri

---

## [Editor Report · Acceptance letter]

PONE-D-25-48903R1

PLOS One

Dear Dr. Thibeau--Sutre,

I'm pleased to inform you that your manuscript has been deemed suitable for publication in PLOS One. Congratulations! Your manuscript is now being handed over to our production team.

Kind regards,

on behalf of

Professor Sheng Du

Academic Editor

PLOS One